# Flax-Derived Carbon: A Highly Durable Electrode Material for Electrochemical Double-Layer Supercapacitors

**DOI:** 10.3390/nano11092229

**Published:** 2021-08-29

**Authors:** Petr Jakubec, Stanislav Bartusek, Josef Jan Dvořáček, Veronika Šedajová, Vojtěch Kupka, Michal Otyepka

**Affiliations:** 1Czech Advanced Technology and Research Institute (CATRIN), Regional Centre of Advanced Technologies and Materials (RCPTM), Palacký University Olomouc, Šlechtitelů 27, 783 71 Olomouc, Czech Republic; veronika.sedajova@upol.cz (V.Š.); vojtech.kupka@upol.cz (V.K.); 2Department of Environmental Protection in Industry, Faculty of Materials Science and Technology, VSB-Technical University of Ostrava, 17. Listopadu 2172/15, 708 00 Ostrava-Poruba, Czech Republic; stanislav.bartusek@vsb.cz (S.B.); jdvoracek@solitaire.cz (J.J.D.); 3Department of Physical Chemistry, Faculty of Science, Palacký University, 17. Listopadu 1192/12, 779 00 Olomouc, Czech Republic

**Keywords:** flax, activated carbon, EDLC, supercapacitor

## Abstract

Owing to their low cost, good performance, and high lifetime stability, activated carbons (ACs) with a large surface area rank among the most popular materials deployed in commercially available electrochemical double-layer (EDLC) capacitors. Here, we report a simple two-step synthetic procedure for the preparation of activated carbon from natural flax. Such ACs possess a very high specific surface area (1649 m^2^ g^–1^) accompanied by a microporous structure with the size of pores below 2 nm. These features are behind the extraordinary electrochemical performance of flax-derived ACs in terms of their high values of specific capacitance (500 F g^–1^ at a current density of 0.25 A g^–1^ in the three-electrode setup and 189 F g^–1^ at a current density of 0.5 A g^–1^ in two-electrode setup.), high-rate stability, and outstanding lifetime capability (85% retention after 150,000 charging/discharging cycles recorded at the high current density of 5 A g^–1^). These findings demonstrate that flax-based ACs have more than competitive potential compared to standard and commercially available activated carbons.

## 1. Introduction

Supercapacitors have been attracting tremendous interest for more than fifty years and are considered an excellent counterpart to batteries for energy storage applications [1]. They can be sorted into two basic classes, based on the charge storage mechanism: electrochemical double-layer (EDLC) or faradic (pseudo) capacitors [2]. Devices based on an EDLC mechanism, where the charge is stored solely between the electrode/electrolyte interface, usually utilize different carbon derivatives such as carbon nanotubes [3], templated carbons [4], graphene [5], and activated carbon [6]. In contrast, pseudo-supercapacitors utilize transition metal oxides [7], conducting polymers [8], MOFs [9], and MXenes [10] as electrode materials, which enable charge storage via fast electrochemical faradic reaction. Both approaches have various advantages that originate from the underlying charge storage mechanism. For instance, EDLC displays a very high power output with respect to pseudo-capacitors. In contrast to pseudo-capacitors, the EDLC-based systems possess lower energy density. Due to the nature of the material, the EDLC-based system exhibits significantly higher lifetime stability than the pseudo-capacitors. This aspect together with the fast charging capabilities and low maintenance costs make EDCL capacitors very suitable for practical applications [11]. Suitable materials for high-performance EDLC capacitors display (i) a sufficiently large specific surface area (SSA) with a suitable pore size distribution; (ii) an interconnected network of pores that enhances the ion transfer and reduces the diffusion influence; (iii) high surface wettability, which improves the pore filling by electrolyte ions; (iv) good electrical conductivity; and (v) cost effectiveness [1,11,12]. Commercially available EDLC capacitors deploy ACs because they meet the abovementioned requirements [13]. Biomass-derived carbons represent an interesting source of ACs since they are abundant and naturally renewable [14]. These bio-derived ACs include, for instance, wood sawdust [15], coffee endocarp [16], rotten tomatoes [17], rice straw [18], winery wastes [19], or sunflower seed shell [20]. However, only a few bio-inspired materials found application in industry [21,22,23,24]. The main drawback of the previously mentioned bio-samples is related to the poor or insufficiently documented lifetime stability of supercapacitors prepared from these materials.

Herein, we present a novel AC-based material prepared from flax (*Linum usitatissimum*). Flax is an abundantly and worldwide available plant [25] finding numerous applications not only in wearable textiles but also in filtration processes, automotive cushioning, acoustic insulation [25], and biomedical applications [26]. Recently, a number of authors have reported the use of activated flax fiber cloth as the electrode platform for supercapacitors [27,28,29,30]. However, all of them used multiple steps or other complicated synthetic procedures, e.g., peculiar control of gas atmosphere [27]. These ACs suffer from either low capacitive performance or low lifetime stability. For instance, high-nitrogen-containing porous carbon fiber sheets prepared from biomass-flax exhibited sufficient lifetime stability only for 5000 cycles [27]. The insufficient lifetime stability was also reported for flax-derived carbon cloth coated with MnO_2_ nanosheets, where authors reported only a 1000-cycle lifetime [28]. Unsatisfactory numbers in terms of both specific capacitance (191 F g^–1^ at 0.1 A g^–1^) and lifetime stability (5000 cycles) were reported for flax fabric modified with carbon nanotubes [30]. In contrast to these previous works, our approach consists of a simple two-step synthesis including a hydrothermal conversion of natural flax and further pyrolytic–chemical activation of the product by potassium hydroxide. This procedure combines the benefits of simplicity, reproducibility, and cost-effectiveness. Such an AC material exhibits a very high SSA (1649 m^2^ g^–1^) accompanied by a microporous structure with the size of pores below 2 nm. All of these facts result in extraordinary electrochemical performance in terms of high values of specific capacitance (500 F g^–1^ at a current density of 0.25 A g^–1^ in a three-electrode setup and 189 F g^–1^ at the current density of 0.5 A g^–1^ in a two-electrode setup), high-rate stability, and outstanding lifetime capability (85% retention after 150,000 charging/discharging cycles recorded at a high current density of 5 A g^–1^). These findings clearly show that a flax-based carbon derivative can be a sustainable source of AC from an abundant plant, which has more than competitive potential in comparison to standard and commercially available activated carbons.

## 2. Materials and Methods

### 2.1. Reagents

Sulfuric acid was purchased from Lach-Ner (p.a. 96%, Neratovice, Czech Republic), and potassium hydroxide (p.a. 90%) and hydrochloric acid (p.a. 36.5%) were purchased from Mach chemikálie (Ostrava-Hrušov, Czech Republic). TimCal Super C45 conductive carbon black powder was obtained from Cambridge Energy Solutions Ltd. (Cambridge, United Kingdom); KetjenBlack EC-600JD was purchased from AkzoNobel (Prague, Czech Republic); and activated carbon YP-80F was purchased from Kuraray Co., Ltd. (Osaka, Japan). All chemicals were used as delivered without any further purification. Ultrapure water (18 MΩ cm^–1^) was used for the preparation of all aqueous electrolyte solutions.

### 2.2. Synthesis of Flax-Derived Carbon

Flax-based carbon was prepared using a two-step procedure: (1) hydrothermal conversion and (2) pyrolytic-chemical activation with potassium hydroxide. In the first step, a 50 g of a flax sample (obtained after fiber processing from the whole plant) was hydrothermally converted to a hydrochar at a pressure of 24 bar and a temperature of 220 °C for 5 h in an in-house designed autoclave. The flax hydrochar (24 g) was rinsed with distilled water and dried at 105 °C for 8 h. This step was followed by the biochar activation process carried out as follows: 24 g of hydrochar was mixed with 72 g of KOH (ratio 1:3), and after complete homogenization, placed in a stainless steel reactor equipped with a corundum crucible. The homogenized sample mixture was then heated to 800 °C at a rate of 12.6 °C min^–1^ using an inert N_2_ atmosphere (99.96%) at a flow rate of 3.2 L min^–1^. Further pyrolysis and activation led to the desired properties. They were influenced by the temperature and pyrolysis time as well as by the addition of the selected activating agent. The set parameters (pyrolysis temperature, pyrolysis time, activating agent) are the result of extensive previous experimentation with samples of different biomaterials. In the case of the flax, the following criteria were proved optimal: the temperature of the final pyrolysis set to 800 °C, the heating time to reach this temperature set to 60 min, and intensive cooling to room temperature set to 45 min. KOH turned out to be the optimum activating agent at a ratio of 1:3 with the sample. Other activating agents and ratios were also tested, but in those cases, lower capacitive properties were achieved. The final carbon sample produced from flax by alkaline fusion takes the form of a glassy mass with high alkalinity. The melt is decomposed and neutralized by a controlled dissolution in 1 mol L^–1^ HCl to a weakly acidic environment while releasing the activated carbon. Further washing with water enabled removal of the acidic component until pH 7 was obtained, which was followed by filtering and drying at 105 °C for 4 h. The mass of the activated material obtained was found to be 4.87 g (i.e., yield ≈ 20%).

### 2.3. Apparatus

High-resolution TEM (HRTEM) images including STEM-HAADF (high-angle annular dark-field imaging) analyses for elemental mapping of the products were acquired with an FEI Titan HRTEM microscope using an operating voltage of 200 kV. For these analyses, a droplet of dispersion of the material in ultrapure H_2_O with a concentration of ≈0.1 mg mL^–1^ was deposited onto a carbon-coated copper grid and dried. An X-ray photoelectron spectroscopy (XPS) measurement was performed on a PHI VersaProbe II (Physical Electronics) spectrometer equipped with an Al Kα source (15 kV, 50 W, spot size 100 μm). The C1s core level of the C-–C bond (value of 284.8 eV) served as a reference value for all binding energies. The MultiPak (Ulvac-PHI, Inc.) software package was used for the evaluation and deconvolution of the data sets obtained. Raman spectra were recorded using a DXR Raman microscope equipped with a diode laser with a 633 nm excitation line. Pore size and surface area analyses (Brunauer, Emmett and Teller, BET) were carried out using a volumetric gas adsorption analyzer (3Flex, Micromeritics) at –196 °C and up to 0.965 bar. The sample was degassed under a high vacuum for 24 h at room temperature using high-purity gasses (N_2_ and He). The BET surface area was determined with respect to Rouquerol criteria [31] for N_2_ isotherm. Pore size distribution (PSD) was analyzed by N_2_-NLDFT 77-carbon slit pore kernel. The model was chosen based on its best goodness-of-fit parameter.

### 2.4. Electrochemical Measurements

A Metrohm Autolab PGSTAT128N instrument (Metrohm Autolab B.V., Utrecht, Netherlands) controlled with the NOVA software package (version 1.11.2) was used for the complex characterization of a flax-derived carbon material in a three-electrode setup. A battery tester BCS-810 (Biologic Company, Seyssinet-Pariset, France) equipped with BT-Lab software (version 1.63) was used for two-electrode cell experiments. Sulfuric acid (*c* = 1 mol L^–1^) was used as a supporting electrolyte in all electrochemical experiments. All measurements were performed at room temperature (22 ± 2 °C).

The three-electrode system comprising a glassy carbon electrode (GCE) was used as the working electrode, a platinum wire electrode served as a counter electrode, and a Ag/AgCl (3 M KCl) electrode served as a reference electrode. Electrochemical impedance spectroscopy (EIS) spectra were recorded using a 10 mV amplitude. All EIS spectra were recorded over the frequency range from 0.1 Hz to 100 kHz at open circuit potential (OCP). The GCE was modified by dropcasting as follows: a 10 µL drop of a powder suspension (2 mg mL^–1^) was coated onto the surface of the GCE working electrode and allowed to dry at ambient temperature to form a thin film.

For the two-electrode system, a symmetrical supercapacitor device was constructed according to the following protocol. Briefly, an active material was homogeneously dispersed in ultrapure water (4.5 mg mL^–1^) and sonicated for 0.5 h. Next, 400 μL of the dispersion was dropcoated onto the surface of a gold disc electrode (diameter 18 mm) and dried in ambient conditions to achieve a mass loading of at least 0.8 mg cm^–2^. For the assembly of the supercapacitor device, two gold disc electrodes with the same loading of the active material were placed in an insulator sleeve (El-Cell insulator sleeve) using Whatman^®^ glass microfiber filter paper with a thickness of 0.26 mm as a separator. The separator membrane was soaked with 100 μL of electrolyte. Stainless steel plungers were used to press the electrodes, and the whole device was tightened and connected.

The specific capacitance in the three-electrode configuration (*C*_s_, F g^−1^) was calculated from galvanostatic charging/discharging (GCD) curves as follows:(1)$$C_{s}=\frac{I\times \Delta t}{m\times \Delta V}$$
where *C**_s_* is the gravimetric capacitance (F g^−1^), *I* is the discharge current (A), Δ*t* is the discharge time (s), Δ*V* is the potential window, and *m* is the mass of the material deposited on the surface of the working electrode (g).

In the two-electrode setup, the specific capacitance of the cell (*C*_s_, F g^−1^) was calculated using Equation (2):(2)$$C_{s}=\frac{4I\times \Delta t}{m\times \Delta V}$$
where *C**_s_* represents the gravimetric capacitance (F g^−1^), *I* is the discharge current (A), Δ*t* is the discharge time (s), Δ*V* is the potential window, and *m* is the total mass of active material on both electrodes. The multiplier of 4 adjusts the capacitance of the cell and the combined mass of two electrodes to the capacitance and mass of a single electrode [32,33]. 

The maximum of energy density and maximum of power density were calculated using the following equations:(3)$$E=\frac{1}{2}\times \frac{C_{s}\times \Delta V^{2}}{4\times 3.6}$$
(4)$$P=\frac{E\times 3.6}{\Delta t}$$
where *C*_s_ is the specific capacitance (F g^–1^) and Δ*V* is the operating window; the dividing factor of 4 is used to normalize the results back to the values of the full-cell (two-electrode) system. The factor of 3.6 converts the energy from joules (the result of Farads × volts) into Wh kg^−1^. Discharging time (s) is represented by Δ*t* [34].

## 3. Results and Discussion

### 3.1. Chemical and Morphological Characterization

The XPS analysis was conducted to unravel the chemical composition of the C-Flax sample. The survey XPS spectrum (Figure 1a) illustrates four predominant peaks of C 1s, N 1s, O 1s, and Fe 2p, located at binding energies 285, 398, 532, and 712 eV, respectively, corresponding to the atomic content of 88.7 at. % for C, 5.7 at. % for O, 2.0 at. % for N, and 0.8 at. % for Fe (inset of Figure 1a). Small amounts of Cu (0.5 at. %) and K (0.3 at. %) elements were also detected. The oxygen content points to the presence of oxygen-containing functional groups, which may contribute to a pseudo-capacitive effect during the electrochemical tests of AC samples. We further analyzed the nature of C atoms via HR-XPS, which showed that the C 1s peak can be deconvoluted into several components (Figure 1b). The dominant peak was located at 284.9 eV and corresponded to sp^2^ carbons. The peak at 286.3 eV corresponded to sp^3^ carbons, other weak peaks located at 287.7 eV and 289.3 eV were attributed to carbons in C–O/C–N and COOH bonds, respectively. The Raman spectrum of the C-Flax sample (Figure 1c) showed two distinct peaks indexed as D-band (1334 cm^–1^) and G-band (1596 cm^–1^). The presence of the first was related to internal structural defects, edge defects, and dandling bonds (i.e., disorder-induced defects of graphitic carbon [35]), and the amorphous nature of carbon [36], while the latter came from the stretching mode of symmetric C–C bonds [37]. The D-band (ID) vs. G-band (IG) ratio was 1.21, which indicated a large number of structural disorders and a partially graphitic structure of this sample. The results from Raman spectroscopy complemented the data obtained by XPS and confirmed the presence of carbon bonds in sp^2^ configuration, which suggested enhanced electrical conductivity of the C-Flax-based system, this being of paramount importance for building EDLC supercapacitors.

Since ACs that are suitable for energy storage applications need a sufficiently high SSA and pore size distribution, these parameters were evaluated by the physisorption analysis. The C-Flax sample exhibited a type I N_2_ adsorption/desorption isotherm (Figure 1d). As evident, the adsorption by the sample occurred at relative pressure below 0.1 and further showed the plateau at higher relative pressures. Such a response indicates a high microporosity of the tested sample [38]. The SSA obtained from BET measurements equaled 1649 m^2^ g^–1^ with a total pore volume of 0.79 cm^3^ g^–1^. As expected, the pore size distribution revealed micropores with a size below 2 nm (Figure 1e). It is worth noting that the C-Flax sample had a high micropore ratio with the V_mi_/V_total_ = 75%, which makes it highly competitive with other porous carbon derived from natural products such as hemp [39], animal bones [40], and argan seed shells [21]. The high SSA together with the microporous structure of C-Flax provides extra space for electrolyte ions, which is crucial for EDLC systems [41,42,43] and therefore predispose C-Flax for supercapacitor applications.

The morphological features of the C-Flax sample obtained from a HRTEM analysis are illustrated in Figure 2. The C-Flax exhibited both flake-like structures with lateral size in micrometer units and small carbon-based agglomerates with the size of particles similar to a flake-like structure (Figure 2a). A selected area electron diffraction (SAED) analysis confirmed the amorphous nature of the C-Flax sample, while the presence of well-developed rings was visible. The energy-dispersive X-ray (EDAX) spectrum of the C-Flax sample obtained from the HRTEM analysis (Figure 2c) depicts dominant peaks for C, N, O, which is in agreement with the XPS results. The dominant element was C accompanied by O and N, which originated from the biomass. The metal elements, i.e., Fe, Cu, Al, K, and other elements such as S may have come from both the technological process of the sample preparation (the activation procedure of the C-Flax sample by KOH contributes significantly to the K content) and from the natural abundance of these elements in flax [44,45]. A darkfield HRTEM image (Figure 2b) was used for EDS chemical mapping of the elements noted above. As shown in Figure 2d, the presence of the particular elements is completely uneven, which can be interpreted by the complex nature of the flax plant, including the presence of different cells of stems, leaves, flowers, etc. It is important to mention that these elements were found in very low abundance, and, therefore, a negligible impact can be expected on the electrochemical performance of the C-Flax sample.

### 3.2. Electrochemical Characterization

The three-electrode arrangement was employed first to test the electrochemical performance of the C-Flax sample. Figure 3a illustrates the cyclic voltammetry (CV) response of C-Flax together with different commercially available activated or conductive carbons such as KetjenBlack, YP-80F (Kuraray) or Timcal. All voltammograms were recorded in the environment of 1 mol L^–1^ sulfuric acid at a constant scan rate of 20 mV s^–1^. The potential window was limited to 1 V because of the thermodynamic stability of water-based electrolytes [46]. The results obtained clearly show that the area under the CV loop of C-Flax against Timcal and KetjenBlack is significantly higher (i.e., 26 and ca. 3 times, respectively at V = 0.3 V), which assumes high conductivity and better capacitative performance of the C-Flax sample. Moreover, it is possible to observe a random drop of current response in different potential window regions (depicted as red dashed rectangles), especially in the case of the KetjenBlack sample. Such a phenomenon suggests poor contact of the tested material with the surface of the working electrode. It is obvious from the CV curves that only the YP-80F (Kuraray) sample derived from coconut shells yielded a current response comparable to C-Flax. The main difference is connected with the stronger redox activity of the YP-80F (Kuraray) sample at around 0.3 V (visualized by a red dashed line). This faradic response is attributed to the redox activity of basic functional oxygen-bearing groups such as carbonyl and quinone, which react with protons (H^+^) in the acidic aqueous electrolytes [47,48,49]. Such a response of the redox reaction may result in higher values of (pseudo)capacitance [50]. However, these pseudo-capacitive effects are responsible for the intrinsic instability of the material, and they are usually accompanied by a decreased value of the lifetime stability [51]. In the case of the C-Flax sample, the CV loop exhibited a blunt and slanted symmetrical shape, indicating the presence of both ESR and EPR resistance [52] and suggesting the prevailing EDLC charge storage mechanism. The quasi-rectangular shape also points to the material with high SSA and microporous structure [53].

As expected, galvanostatic charging/discharging (GCD) curves resemble results obtained from the CV experiments. Thus, minor deflection from ideal behavior in terms of symmetric quasi-triangular shapes was observed (Figure 3c). Figure 3d shows the specific capacitance dependency on current density recorded for three independent working electrodes modified with a C-Flax sample. As can be seen, a maximum specific capacitance of 500 F g^–1^, as an average value from three measurements, was obtained at a current density of 0.25 A g^–1^. It should be kept in mind that higher standard deviation (13.4%) received at a current density of 0.25 A g^–1^ can be explained by the structure of the C-Flax sample. The sheets with different dimensions possess different sizes of SSA for the adsorption of ions. Since the three-electrode setup utilizes a significantly lower mass of the tested material, a difference in specific capacitance can be expected, especially at low current densities. The higher drop of specific capacitance observed from the change of current density from 0.25 to 0.5 A g^–1^ is because of the porosity of the C-Flax sample. As discussed earlier, the BET measurement revealed the size of SSA 1649 m^2^ g^–1^ with almost uniform distribution of pores below 2 nm in size. It indicates that at very low current densities, hydrated ions from sulfuric acid electrolyte (2.8 Å for H^+^ and 3.8 Å for SO_4_^2–^) [54] have more time to access the entire available space, which results in higher values of specific capacitance. The results acquired from GCD experiments illustrate the high potential of the C-Flax sample in supercapacitor applications since such materials show significantly better values of specific capacitance than similar systems derived from different biomass precursors (Appendix A).

Figure 3e illustrates the electrochemical impedance spectrum (EIS) of the C-Flax sample (magnified version is visualized in Figure 3g) in the form of a Nyquist plot. The EIS response was modeled using the modified de Levie’s transmission line [55] (combination of resistive and capacitive elements; Figure 3f). The spectrum obtained clearly describes the following facts: (1) the absence of semicircle in the high-frequency region denotes the lack of faradic reaction [56]. Such an observation is in good agreement with the theory that carbon porous materials should never show a semicircle in the high-frequency region since the charge storage mechanism is driven only in a physical manner [57]. (2) In the region of middle frequencies (diffusion-controlled Warburg impedance), a 45° line was observed, proving the high porosity of the C-Flax sample. Two main factors have an influence on the size of this region: the speed of the adsorption process and the speed of the diffusion of ions within the interconnected porous structure. With higher loadings, the diffusion in the material slackens, and, thus, the region of middle frequencies is expanded [58,59,60]. (3) In the region of the low frequencies, a steep increase in the imaginary part of the impedance spectrum was observed. It reflects the fact that all reactive sites are fully accessible within a short time, which results in capacitor-like behavior [57,61]. The deviation from the theoretical 90° line can be observed, due to the interfacial impedance that is typical for the porous electrode materials [57].

The symmetrical two-electrode capacitor system based on the C-Flax sample was built for practical comparative reasons. In this setup, we used the water-based suspension of the C-Flax sample without any additives (conductive carbons or binders) to avoid the parasitic contribution of these substances to the overall electrochemical performance. The GCD response of the C-Flax sample recorded within the range of current densities from 0.5 to 20 A g^–1^ is illustrated in Figure 4a,b. As expected, the shape of GCD curves became more symmetrical, suggesting the improved adsorption of ions at the carbon surface, compared to the three-electrode setup. As demonstrated in Figure 4c, a maximum specific capacitance of 189 F g^–1^ was achieved at a current density of 0.5 A g^–1^. The resulting specific capacitance vs. current density profile (Figure 4c) reflects the outstanding properties of the C-Flax sample. Such a statement is evidence by the minimum difference in C_S_ up to 5 A g^–1^. A small decrease in specific capacitance was observable only at very high current densities (>10 A g^–1^), suggesting high stability of the C-Flax sample. As illustrated in Figure 4c, the values of C_S_ were reduced approximately threefold, compared to the three-electrode setup. Such an observation is in good agreement with the statement that commercial supercapacitors containing activated carbons account only for one-third of the total device weight because of the presence of inactive yet indispensable components such as current collectors, electrolyte, separator, or package [62]. For this reason, material-based results (those obtained from a three-electrode setup) should be reduced by a factor of 3–4 to arrange a realistic estimation of device-level values [57,63]. A rate test (10 cycles) ranging from 0.5 to 20 A g^–1^ proved the remarkable stability of the C-Flax sample, as shown in Figure 4d. Figure 4e shows the lifetime cycling performance of the C-Flax-based supercapacitor operating in the sulfuric acid electrolyte at a high constant current density of 5 A g^–1^. It is evident that the capacitance response was retained at 94.5% after 30,000 cycles and decreased very slowly up to 150,000 (capacitance retention 85%). The inset of Figure 4e depicts the first and last five cycles for comparative reasons. Such results are significantly better than those obtained from similar carbon-based materials such as sugar cane bagasse [64], sunflower stalk [65], rice husk [66], or coconut waste [67]. The ability of the C-Flax sample to serve as a material suitable for supercapacitor application is also illustrated by the Ragone plot (Figure 4f). The energy density up to 6.58 Wh kg^–1^ (based on the mass of active material) was achieved at the lowest value of current density (0.5 A g^–1^). With increasing power density, the benefit of the microporous structure is reflected in the stable performance of the C-Flax system, which retained 4.53 Wh kg^–1^ at 10,000 W kg^–1^. Since both energy and power density values are inadequately reported for similar materials (see Appendix A), commercially available activated carbon YP-80F (Kuraray company) derived from coconut shells was tested for comparative reasons. Under the same experimental conditions, we found that the YP-80F sample exhibited significantly lower performance in terms of energy density (E = 4.6 Wh kg^–1^ at 0.5 A g^–1^), as shown in Figure 4f. Moreover, it was not possible to obtain acceptable results at current densities higher than 5 A g^–1^. As evident, the C-Flax material exhibits remarkable electrochemical performance in terms of specific capacitance, rate stability, and lifetime cycling stability. Such material provides an outstanding alternative to commercially available activated carbons.

## 4. Conclusions

A flax-based carbon derivative was prepared by a simple two-step synthesis including hydrothermal conversion of natural flax and further pyrolytic-chemical activation of the product by potassium hydroxide. Prepared activated carbon material exhibited high surface area (1649 m^2^ g^−1^) and high micropore ratio (V_mi_/V_total_ = 75%), which makes it competitive with other porous-carbon-derived materials. Most importantly, the activated carbon material exhibited an outstanding electrochemical performance in terms of high values of specific capacitance (500 F g^–1^ at a current density of 0.25 A g^–1^ in the three-electrode setup and 189 F g^–1^ at a current density of 0.5 A g^–1^ in the two-electrode setup.), high rate stability, and outstanding lifetime capability (85% retention after 150,000 charging/discharging cycles recorded at a high current density of 5 A g^–1^). These findings clearly show that flax-based carbon derivatives offer a sustainable source of AC from an abundant plant, which makes them more than competitive with standard and commercially available activated carbons.

## Figures and Tables

**Figure 1 nanomaterials-11-02229-f001:**
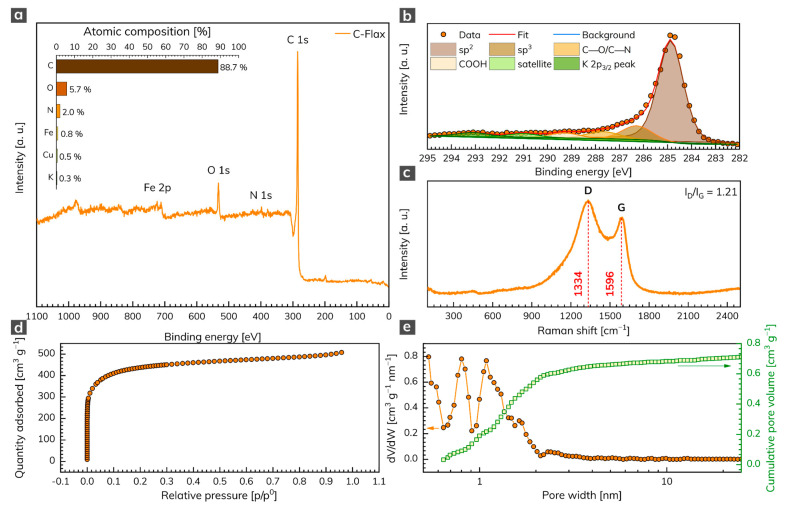
(**a**) XPS survey spectrum of C 1s, N 1s, O 1s, and Fe 2p with related atomic % weight contributions (inset). (**b**) Deconvoluted C 1s peak. (**c**) Raman spectrum of C-Flax sample. (**d**) Adsorption/desorption isotherm of C-Flax sample and its (**e**) related pore size distribution and cumulative pore volume.

**Figure 2 nanomaterials-11-02229-f002:**
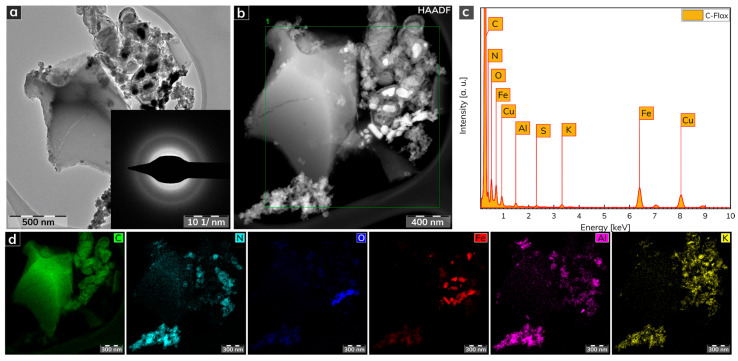
(**a**) HRTEM image of a C-Flax sample. (**b**) Darkfield HRTEM image used for EDS chemical mapping. (**c**) EDAX spectrum of C-Flax sample. (**d**) EDS elemental mapping of C-Flax sample, including carbon, nitrogen, oxygen, iron, aluminum, and potassium atoms. Scale bars were unified from the graphical point-of-view to provide better visibility.

**Figure 3 nanomaterials-11-02229-f003:**
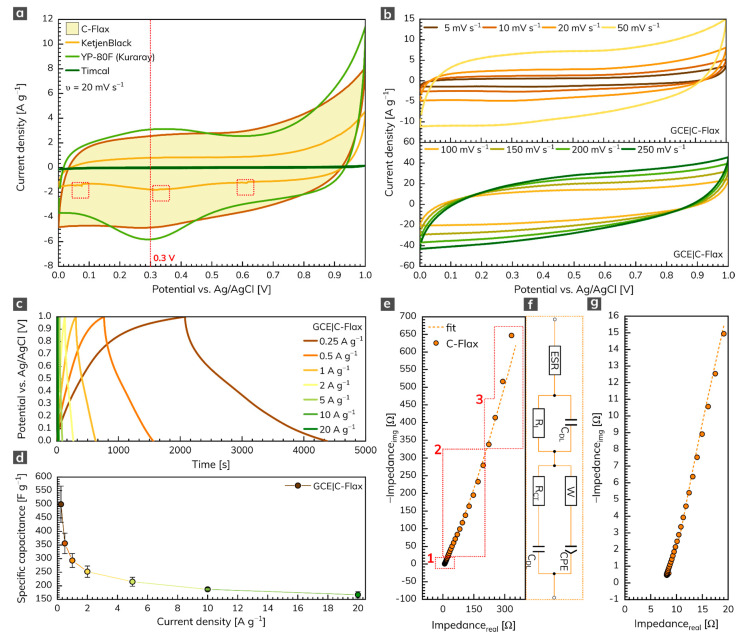
(**a**) CVs of C-Flax and different commercially available samples recorded at a constant scan rate of 20 mV s^–1^. (**b**) CVs of C-Flax sample recorded at scan rates ranging from 5 to 250 mV s^–1^. (**c**) GCD curves of C-Flax sample recorded at different current densities and (**d**) corresponding specific capacitance vs. current density profile. (**e**) Nyquist plot of the C-Flax sample, its magnified version (**g**), and electrical circuit used for data fitting (**f**). All experiments were performed in 1 mol L^–1^ sulfuric acid.

**Figure 4 nanomaterials-11-02229-f004:**
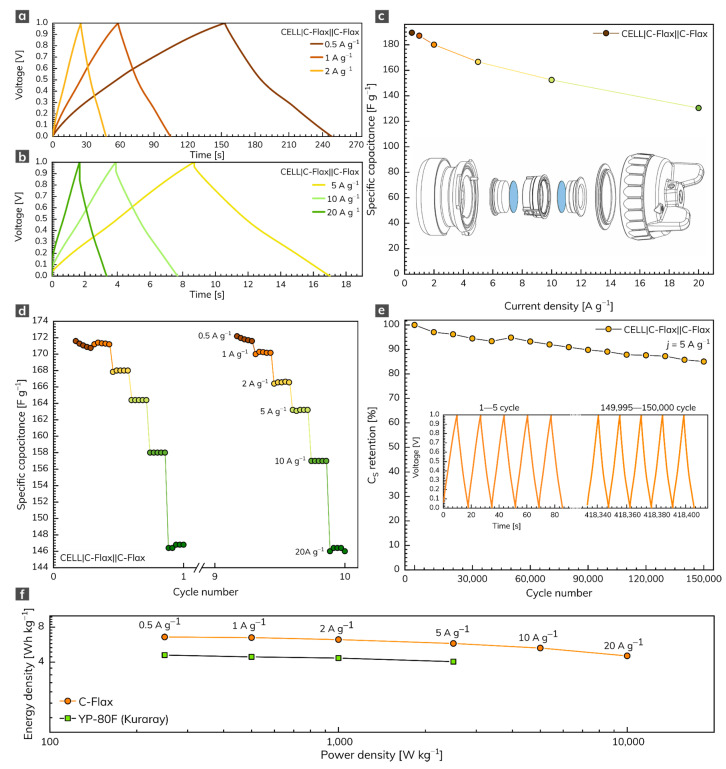
(**a**,**b**) GCD profiles of C-Flax material recorded at different current densities and (**c**) corresponding specific capacitance vs. current density profile. (**d**) Rate-test of C-Flax sample recorded at current densities ranging from 0.5 to 20 A g^−1^. (**e**) Specific capacitance retention of C-Flax sample after 150,000 GCD cycles. (**f**) Ragone plot of C-Flax and YP-80F (Kuraray) sample. All experiments were performed in 1 mol L^–1^ sulfuric acid.

## Data Availability

All data are available upon reasonable request.

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
