# Peer review of "Flax-Derived Carbon: A Highly Durable Electrode Material for Electrochemical Double-Layer Supercapacitors"

_nanomaterials, 2021, doi:10.3390/nano11092229_

Round 1

Reviewer 1 Report

In this manuscript, the author reported two-step synthetic procedure for the preparation of activated carbon from natural flax, and they used the as-prepared flax-derived AC as supercapacitor electrodes. Overall, this work is of interest to the biomass-derived carbon and energy storage field. However, here are still the following problems need to be addressed by the authors:

  1. In the abstract, almost half of the abstract seems to be introduction, so the “Abstract” should be refined.
  2. Fig.2c, EDS spectrum, why are there so many kinds of metal elements? The EDS result can not be the sole evidence for the presence of elements in carbon materials, it is not credible. Other evidences should be provided.
  3. Fig.3c and 3d, as the current density is less than 0.5 A g-1, especially 0.25 A g-1, the GCD curves deviate seriously from the triangle curves, so the calculated specific capacitance is not reliable. In Fig.3d, the maximum current density is only 5 A g-1, it should be at least increased to 10 A g-1.
  4. Fig.3e, the EIS plots are not similar to the most supercapacitor electrode, they should be measured again and explain why.
  5. There are several typos and the English might be substantially improved in all parts of the manuscript.

Reviewer 2 Report

In this work, the authors employed two-step activation method to prepared activated carbon for supercapacitor application. The innovativeness of this paper needs to be clarified.

(1) Long sentences throughout this paper are overused and need to be corrected, and the grammar need be improved.

(2) Lines 46-47 “This aspect makes EDCL capacitors more suitable for practical applications.” It is not just the high life-time stability that needs to be considered in the practical application, there are many other aspects to consider such as specific capacitance etc.

(3) Lines 47-52 this sentence should be improved.

(4) Lines 95-96, can the ACs reach the pH of 7.0 by washing with HCl?

(5) Why did this article apply hydrothermal method as a pretreatment, which was not shown in introduction?

(6) Why are the Fe and Cu presented in XPS, which are not involved in the preparation process?

(7) Lines 175-184: please provide the detailed information for processing pore diameter distribution. When using different models (i.e., slit-stand model, etc.), the pore diameter distribution will give different trend.

(8) In this paper, what is the basis for selecting the pyrolysis temperature and the hydrothermal condition? Why the ratio of raw material to KOH is 1:3, what is the holding time after the pyrolysis temperature reaches 800℃? None of these make a clear statement.

(9) The atomic composition of XPS requires further verification.

(10) Line 229 “As expected, galvanostatic charging/discharging (GCD) curves resemble results obtained from the CV experiments.” The Cs of the material needs to be calculated by CV curves. You can refer to the following literature:

“Zhou Liao., Hong-Yu Su., Jie Cheng., Guo-Tao Sun., Lin Zhu., Ming-Qiang Zhu., CoFe2O4-mesoporous carbons derived from Eucommia ulmoides wood for supercapacitors: Comparison of two activation method and composite carbons material synthesis method, Ind. Crop. Prod.2021;171:113861.”.

(11) In conclusions, in addition to the electrochemical properties, the pore size and specific surface area of the material need to be elaborated.

Round 2

Reviewer 2 Report

Accept